

# Sixty years of change in avian communities of the Pacific Northwest

Jenna R. Curtis and W. Douglas Robinson

Department of Fisheries & Wildlife, Oregon State University, Corvallis, OR, United States

## ABSTRACT

Bird communities are influenced by local and regional processes. The degree to which communities are dynamic has implications for projecting responses in community composition as birds track geographic shifts of their habitats. Historic datasets offer a legacy of information that can be used to quantify changes over time in avian community composition. A rare, highly-detailed avian survey of multiple habitat types in the Willamette Valley, Oregon, was conducted in 1952. We resurveyed the same sites in 2013 and evaluated whether observed results agreed with theoretical patterns of community change. We compared alpha, beta, and gamma diversity between survey periods and evaluated shifts in categorical abundances of species. Most patterns of change were consistent with community turnover. Nearly 50% of species were replaced over six decades, with increased species richness and decreased evenness at local and regional spatial extents. Patterns of regional species turnover reflected local turnover. Evidence that local shifts in habitat type drove bird community change were not strongly supported, although historic data on habitats within study plots were limited to macro-level aerial photographs. Thus, regional factors and structural changes likely played important roles determining species composition and abundance.

Corresponding author
Jenna R. Curtis,
jenna.curtis@oregonstate.edu

## INTRODUCTION

Understanding the long-term consequences of environmental variation on biodiversity is central to the conservation of ecosystems (*Vitousek et al., 1997*). Birds are especially responsive to environmental variation (*Temple & Wiens, 1989*; *Crick, 2004*). Change over time in species richness and patterns of abundance within local bird communities can be indicative of environmental change (*Vale, Parker & Parker, 1982*; *Root, 1988*; *Crick, 2004*), local habitat variation (*Knick & Rotenberry, 2000*; *Brown et al., 2001*; *Rotenberry & Wiens, 2009*), or immigration and extinction at a regional level (*Ricklefs, 1987*; *Loreau & Mouquet, 1999*). Both habitat characteristics and regional species assemblages influence local species communities, but the degree to which such relationships are dynamic over decades is not well understood. Detailed, long-term data capable of quantifying change within local communities are rare. Yet such data are necessary to empirically test predictions of community change (*Igl & Johnson, 2005*; *Tingley & Beissinger, 2009*).

Previous studies suggest several general, non-exclusive ways in which communities can change, particularly in systems experiencing increasing anthropogenic pressure (*Catterall et al., 2010*; *Shultz, Tingley & Bowie, 2012*). Avian communities are generally expected to vary over time, as birds respond quickly to environmental change (*Temple & Wiens, 1989*). However, some communities may express more long-term stability than others. Biologically diverse communities may be more stable, or resilient to change in the face of species invasion or population fluctuations, than simpler communities (*MacArthur, 1955*; *Elton, 1958*). While constancy in communities is unexpected, the extent to which communities change over time may be influenced by community composition, structure, or diversity (*McCann, 2000*).

Community diversity may decrease over time in cases where the number of species capable of inhabiting an area is reduced, usually in conjunction with habitat loss, anthropogenic disturbance or urbanization (*Beissinger & Osborne, 1982*; *Strohbach, Hrycyna & Warren, 2014*). This pattern has been referred to as "diversity decay" (*Catterall et al., 2010*). Homogenization of communities may also occur when an influx of invasive, generalist species across habitats reduces diversity between areas, skewing abundances in favor of common invaders (*McKinney & Lockwood, 1999*; *Davey et al., 2012*). Though invasion and homogenization can temporarily increase overall species richness, a pattern of systematic increase in biodiversity is rarely posited in scientific literature, despite evidence some urban environments support higher species diversity (*Marzluff, 2014*).

Fluctuations in individual abundances may result in some species going locally extinct as other species colonize. This pattern of "species turnover" produces changes in community composition over time without significant changes in species richness (*Diamond, 1969*; *Parody, Cuthbert & Decker, 2001*; *Catterall et al., 2010*; *Shultz, Tingley & Bowie, 2012*). Pronounced turnover of biological systems has been observed across the globe (*Dornelas et al., 2014*), amidst range shifts and species losses (*Vitousek et al., 1997*; *Thomas & Lennon, 1999*). Yet there remains disagreement as to which of these patterns is likely to describe long-term avian community change at smaller spatial extents. Long-term avian surveys on multiple geographic extents, coupled with assessments of habitat change, may provide the best opportunity to test the applicability of these different models in a variety of systems.

We resurveyed sites from a 60-year-old historic dataset. This dataset is uniquely valuable because raw count data from each survey of each site were published in Richard Eddy's (*1953*) master's thesis. Our objective was to evaluate the nature of avian community change in the Willamette Valley, Oregon, and to characterize long-term variation in avian diversity between and among different habitats. We compared alpha, beta, and gamma diversity, as well as species turnover and categorical abundance levels. Changes in land use and habitat were measured using aerial photographs and satellite imagery. Breeding Bird Survey data from western Oregon were used to assess to what degree local differences in alpha and beta diversity between eras agreed with regional gamma diversity. Overall, we evaluated whether observed results agreed with any theoretical patterns of community change.

## METHODS

### Historic avian surveys

In 1952, Richard Eddy surveyed birds at 6 sites within 50 km of Corvallis, Oregon (*Eddy, 1953*). Sites ranged in size between 8 and 20 hectares. Eddy non-randomly selected sites to represent 6 habitat types: coniferous, oak woodland, marsh, mixed deciduous, riverine/riparian, and "brushy". The objective of Eddy's study was to characterize the summer bird community around Corvallis (*Eddy, 1953*). His thesis was the first to provide quantitative information on the abundances of birds in the area. He did not measure vegetation, but qualitatively described presence of dominant tree and shrub species. Eddy recorded the number and species of birds observed by walking through a given site for 2 h between 0500 and 1000. He repeated this method 10 times for each site, visiting each site every 2–13 days in a non-systematic fashion starting the second week of June through August 24 (*Eddy, 1953*). Two sites were occasionally visited within a single day. *Eddy (1953)* reported using 8 × 25 power binoculars to observe birds. Because many commonly heard but rarely seen species such as Pacific Wren and Hermit Warbler were absent from his lists, we suspect Eddy relied mostly, if not entirely, upon visual detections (*Curtis, 2015*).

### Modern surveys

Eddy did not provide maps of his survey sites. We relocated each historic site as accurately as possible using Eddy's site descriptions and aerial photographs of Benton County from 1956 (*US Department of Agriculture, Farm Service Agency, 1956*). We could not relocate the "brushy" site based on his descriptions, so did not resurvey that habitat. For two sites, his descriptions were sufficient to establish the general location but not the actual plot boundaries. We identified "likely areas" adjacent to the selected survey areas that contained the same habitat types as our plots. These surrounding areas were akin to buffers around the plots we surveyed and could include some of the habitat surveyed by Eddy in 1952. We compared bird communities in the 2 sites to adjacent "likely areas" to confirm site placement did not affect survey results (Appendix S1).

We used ArcGIS (*ESRI, 2013*) to designate site boundaries and overlaid a 200 m square grid aligned with the longest axis of each site. We spot-mapped each of the 5 relocated sites 5 times during the 2013 breeding season beginning mid-May and ending the first week of July. We used the spot mapping protocol described in *Bibby et al. (2000)*. Beginning within 10 min of sunrise, we systematically walked across each site from grid point to grid point until the entire area was surveyed. Spot mapping replicates Richard Eddy's area search methods and collects identical information, but additionally records location and observation data for every detection within the site. We recorded the geographic locations, species, sex, number of individuals, detection method (visual vs. auditory), and any relevant territorial or breeding behavior of all birds encountered during a survey period. We did not conduct surveys on days with heavy rain.

Our objective was to survey birds during the breeding season when detectability is highest and local populations of breeding adults are least influenced by non-breeding birds moving through the landscape (*Ralph, Sauer & Droege, 1998*). Detectability of

many species may decline after breeding activities cease. Most breeding activity in Oregon occurs from mid-May through early July (*Marshall, Hunter & Contreras, 2003*). Eddy surveyed birds from June through August. It was necessary to restrict data comparisons to Eddy's first 5 visits to each site because his last 5 visits fell outside the primary breeding season. Detections from the last 5 visits in Eddy's dataset were likely to be affected by reduced detectability of non-singing local breeding birds, as well as the presence of passage migrants or post-fledging family groups. We evaluated the effects of reducing the number of visits and found they were minimal (*Curtis, 2015*). Because abundances and detectability remain generally stable throughout the breeding season (*Ralph, Sauer & Droege, 1998*), it is unlikely beginning modern surveys 2 weeks earlier than historic surveys unduly influenced our results.

## Regional breeding bird surveys

To address questions regarding the spatial extent of changes observed in bird communities, we evaluated patterns of community change on a regional level. The regional avian community was defined by 10 Breeding Bird Survey (BBS) routes within the Willamette and northern Umpqua valleys selected for their similarity to geographic features and habitat surrounding Corvallis, Oregon. Routes ranged between 17 and 134 km of Corvallis. These routes (and BBS route number) were: Tualatin (002), Umpqua (018), Days Creek (026), Adair (033), Scio (034), Dayton (040), Elkton (050), Canby (202), Salem (237), and Lorane (243). Individual route data was downloaded from the USGS Patuxent Wildlife Research Center FTP site (*Pardieck, Ziolkowski Jr, Hudson, 2014*). All years of available data between 1966 and 2012 for a given route were used for analysis.

## Environmental traits

To quantify changes in land use and vegetation cover between historic and modern surveys, we scanned high-resolution digital images of 1956 aerial photographs from Benton County (*US Department of Agriculture, Farm Service Agency, 1956*). We overlaid images onto a 1 m resolution satellite photograph of Oregon from 2012 (*US Department of Agriculture, Farm Service Agency, 2012*). We visually classified all habitat cover within 150 m of each site for both survey eras based on observable physical characteristics of the vegetation. We drew freehand polygons to mark the boundaries between vegetation classes. This approach circumvented the limitations associated with computer-based modelling and assigning land cover classes over pre-established pixel cells (*US Geological Survey, 2012*). Each polygon was classified down to the Macrogroup level using the US National Vegetation classification system (*US Geological Survey, 2012*). For cases where the vegetation class for a given area was unclear, landscape data from the US Gap Analysis Project (*US Geological Survey, 2012*) was consulted to help determine the most likely classification for that polygon. Mean elevation, area in square meters, and percent cover of each land use and vegetation classification for each site were calculated using ArcGIS (*ESRI, 2013*). We compared percent cover values between years to quantify changes in habitat cover for each site.

## Statistical analysis

All statistical analyses were performed in program R (*R Core Team, 2013*). Eddy's use of only visual detections was realized after an initial comparison of the datasets. It was necessary to remove non-visual detections from the modern data to make them comparable to the historic dataset. Without non-visual detections, raw counts of individuals from the modern data were inaccurate because many individual birds were identified aurally and no effort was expended to visually confirm identifications. Therefore, when applicable, we used both datasets including and omitting non-visual detections for statistical analyses. Species richness estimates, accumulation curves, and diversity indices were obtained using the package "vegan" (*Oksanen et al., 2013*).

### Species diversity comparisons

To account for undetected species, we estimated actual species richness using Chao's first estimator ("Chao 1"; *Chao, 1984*). Chao 1 estimates richness based on number of observed species and number of species seen only once or twice during a survey period.

Alpha diversity is a metric of site-specific species variety. We limited our alpha diversity index analyses to birds detected at least twice during a survey year across all sites ($n = 102$ all detections, 91 visual only). Removing the rarest species (singletons detected only once during our modern surveys) reduced noise in the data and eliminated species that were not using the sites for breeding. We calculated compound alpha diversity for total number of individuals of each species observed at a given site during a survey era using the inverse Simpson's diversity (1/D) index. Simpson's inverse is the reciprocal of the chance that two randomly-sampled individuals will be of the same species. 95% confidence intervals for inverse Simpson's diversity were obtained by percentile bootstrapping data from a given site and survey period for 9,999 iterations (*Hammer, Harper & Ryan, 2001*). Inverse Simpson's indices between survey periods were compared using paired *t*-tests under the null hypothesis that the modern community diversity of a site was not significantly different from the historic diversity (*Brower, Zar & Von Ende, 1998*; *Hammer, Harper & Ryan, 2001*).

Beta diversity uses metrics of dissimilarity to investigate differences between multiple communities separated by space and/or time (*Anderson et al., 2011*; *Chase et al., 2011*). We quantified beta diversity using the modified Raup-Crick method ($\beta_{RC}$; *Chase et al., 2011*). $\beta_{RC}$ is independent of changes in alpha diversity and does not depend on the number of species within each community (*Anderson et al., 2011*). This approach evaluates whether pairs of communities for a given time period are more or less different than chance. Calculations are based on the number of species in each site and in the regional pool, as well as the proportion of sites occupied by each species. We calculated beta diversity using the program R code provided by *Chase et al. (2011)*.

To identify significant differences in beta diversity among non-random pairs of communities, it was necessary to compare against null communities generated by chance. To derive null communities, we randomly sampled a number of species from the entire species pool equal to the number of species for a given pair of sites. We estimated the

probability that the observed number of shared species in a pair of communities was equal to or lower than the null expectation. This probability was re-scaled to a metric ranging from −1 to 1, where communities with lower values are less dissimilar than expected, and communities with higher values are more dissimilar than expected (*Chase et al., 2011*). To compare beta diversity for communities between years, we calculated pairwise dissimilarity matrices for all sites within a given year then tested for differences in mean dissimilarity values using paired 2-sample *t*-tests.

Turnover represents the instability of a species pool over time. Many measures of turnover fail to account for non-detected species that may immigrate or go extinct from the local species pool between years (*Boulinier et al., 1998*). For this study, species turnover was defined as the complement of the estimated total number of species shared between 2 time periods (*i* and *j*) conditioned on the estimated total number of species during time *j* (*Nichols et al., 1998*). The number of shared species was estimated with the Chao 1 richness estimator using the abundance data from time *i* only for species also detected during time *j*. Because Chao 1 approximates actual species richness, including an estimate of species missed by the surveyor, the result is a conservative estimate of shared species richness that accounts for non-detected species. This value was divided by Chao's estimate of species richness for time *j* to produce the probability that any given species at time *j* was a species present during time *i*. The complement of this was the estimated probability that a species is "new", or not present during the initial surveys. We calculated turnover for each site as well as the entire study area between survey years. Estimates of standard error were obtained by standard nonparametric bootstrapping of the data for 1,000 iterations using the "boot" package in program R (*Canty & Ripley, 2014*).

When examining species turnover on a regional level, it was necessary to use Chao's estimate rather than observed richness due to the structure of regional BBS data. The estimated number of shared species across all BBS routes was frequently as large as or larger than the observed richness for the subsequent year. To reduce bias, the denominator on which the estimate of richness is conditioned must be representative of the relevant species pool (*Cam et al., 2000*). Because the area represented by the regional species pool was large, there was a considerable discrepancy between observed and estimated species richness. Conditioning upon estimated regional species richness for the second survey period produced a less biased estimate of turnover compared to observed richness, and was more appropriate for examining regional gamma diversity.

To investigate changes in gamma—or regional—diversity, we first defined the regional species pool from which immigrations into the local sites might occur. Because the number of BBS routes surveyed varied across years, we used individual-based rarefaction to generate subsamples for each year after standardizing for differences in survey effort (*Gotelli & Colwell, 2001*). Rarefaction was performed on all routes from a given year for 1,000 iterations using package "vegan" in Program R (*Oksanen et al., 2013*). Mean species richness was then estimated for the rarefied samples from each year using Chao 1 (*Chao, 1984*). We calculated mean yearly regional species turnover from the rarefied samples using a method similar to local turnover. Durbin-Watson tests were used to test

for autocorrelative structure in the data (*Durbin & Watson, 1950*; *Durbin & Watson, 1951*). When serial correlation was present, we used linear filtering to adjust the data prior to fitting regression lines and estimating trends.

### Changes in abundance

It was necessary to account for lack of direct comparability between historic and modern estimates of abundance due to differences in observation methods. To detect large shifts in abundances, we organized species into categorical levels of abundance. We based categories on the mean number of individuals per visit to a site for a given year. Species were classified as follows (mean number of individuals per visit provided in parentheses): "rare" (up to 1.5 individuals per visit), "uncommon" (between 1.5 and 4.5 individuals per visit), "common" (between 4.5 and 10 individuals per visit), and "abundant" (over 10 individuals per visit). Species with zero individuals per visit at a given site during one of the survey eras were classified as "not detected" for that era.

We categorized changes in abundance between years. Trend classifications ranged from "strongly declining" to "strongly increasing" based on both the direction and magnitude of the shift in abundance category. Species that retained their historic categorical abundance classification were classified as "no change". We tested for shifts in the distribution of species among abundance categories across all sites between years using chi-square tests under the null hypothesis that the overall distribution of species within categories was not different between 1952 and 2013. For comparisons of abundance distributions at individual sites, we used Fisher's exact tests because the assumptions of the chi-square test were not met. For species strongly increasing or decreasing locally, we considered possible ecological explanations and compared our results to regional BBS population trends. Trends were obtained from *Sauer et al. (2014)* who use a hierarchical model to calculate changes in annual indices of abundance.

## RESULTS

### Land use and vegetation cover

To determine if environmental conditions at survey sites changed, we quantified differences in the percent cover of multiple vegetation and land use classes. The amount and direction of habitat change varied among sites (Figs. S1–S5). The Willamette River site experienced some of the greatest overall changes in habitat percent cover. Historically this site was nearly 50% wetland and approximately 30% open water. By 2013, floodplain forest dominated the site with 85% cover. Canopy in the burned portion (about 15%) of the mature coniferous site closed, but was otherwise similar in structure. Both the mixed deciduous and oak woodland sites experienced some canopy closure. The mixed deciduous site changed from primarily riparian forest to a mixed forest/pasture habitat with nearly 50% deciduous cover. Similarly, the oak woodland site lost almost 15% deciduous cover to expanding coniferous forest. *Eddy (1953)* did not quantify understory vegetation so we could not quantify changes in habitat structure. In conjunction with an overall increase in coniferous canopy cover, the three forest habitats remained forested, while the Willamette River floodplain site increased in forest cover. The marsh site was superficially very similar,

**Table 1  Observed and Chao 1 estimated number of species (S).** Richness calculated across all sites and by individual sites. Values for the modern data were calculated for the entire dataset as well as for the data after removing non-visual detections to better replicate historic survey methods. 95% confidence intervals for Chao's estimated number of species are provided in parentheses.

| Site | Richness measure | 1952 | 2013 (all detections) | 2013 (visual only) |
|---|---|---|---|---|
| Overall | Observed S | 79 | 101 | 85 |
| | Chao 1 S | 87 (79–99) | 116 (101–134) | 109 (85–137) |
| Coniferous | Observed S | 32 | 36 | 19 |
| | Chao 1 S | 33 (32–40) | 46 (36–78) | 24 (19–58) |
| Marsh | Observed S | 34 | 61 | 51 |
| | Chao 1 S | 35 (34–40) | 69 (61–83)[a] | 55 (51–62)[a] |
| Mixed deciduous | Observed S | 32 | 43 | 36 |
| | Chao 1 S | 33 (32–36) | 46 (43–53)[a] | 42 (36–57) |
| Oak Woodland | Observed S | 25 | 52 | 31 |
| | Chao 1 S | 25 (25–28) | 58 (52–72)[a] | 37 (31–51)[a] |
| Willamette | Observed S | 27 | 46 | 36 |
| | Chao 1 S | 29 (27–43) | 48 (46–53)[a] | 39 (36–45) |

**Notes.**

[a] Indicates a significant increase in estimated richness compared to historic values.

but Eddy's description indicates the marsh was open to cattle grazing, which has now been eliminated.

## Species diversity

Stability and turnover predict species richness will remain the same, while diversity decay predicts richness will decrease. Observed species richness was higher in modern study sites compared to the historic surveys when aural detections were included (Table 1). Thirteen rare species were removed from historic dataset and 18 from the modern dataset, 14 of which were visually detected. After removing non-visual detections, observed species richness remained an average of 4.8 species higher than historic richness, except for the coniferous site, which decreased in richness. Estimates of species richness accounting for rare and undetected species showed significant increases in richness for the marsh and oak woodland sites.

Alpha diversity is linearly related to species richness, and should exhibit a similar response as richness under each pattern of community change. After removing rare species, the inverse of Simpson's diversity was an average of 4.8 higher for modern sites (Table 2). Across all sites, alpha diversity experienced a statistically significant increase between years (all detections: $t_{3905} = 19.3$, $p < 0.001$; visual detections: $t_{4379} = 16.2$, $p < 0.001$). The only significant site-specific increase occurred at the marsh site (all detections: $t_{2579} = 23$, $p < 0.001$; visual detections: $t_{2832} = 20.9$, $p < 0.001$). The coniferous and oak woodland sites both showed significant decreases in diversity considering only visual detections (coniferous: $t_{298} = -6.6$, $p < 0.001$; oak woodland: $t_{410} = -3.8$, $p < 0.001$). There was no evidence of changes in diversity for the mixed deciduous and Willamette River sites. Removing non-visual detections had the general effect of decreasing modern alpha
**Table 2 Simpson's diversity values (1/D) for historic and modern survey eras.** 95% confidence intervals are provided in parentheses.

| Site | 1952 | 2013 (all detections) | 2013 (visual only) |
|---|---|---|---|
| Overall | 9.28 (8.66–9.93) | 30.90 (29.62–31.91)[a] | 23.47 (22.04–24.58)[a] |
| Coniferous | 15.93 (13.51–17.33) | 13.95 (12.64–14.93) | 7.15 (6.01–8.10)[a] |
| Marsh | 3.14 (2.97–3.33) | 10.89 (10.14–11.66)[a] | 9.63 (8.84–10.42)[a] |
| Mixed deciduous | 12.24 (10.51–13.66) | 12.92 (11.44–14.26) | 11.11 (9.45–12.5) |
| Oak | 11.12 (9.63–12.21) | 12.19 (10.67–13.7) | 7.55 (6.54–8.6)[a] |
| Willamette | 12.69 (11.34–13.41) | 12.08 (10.93–13.14) | 10.73 (9.15–12.06) |

**Notes.**

[a] Indicates significantly different modern diversity compared to historic values (Paired $t$-test, $p < 0.05$).

**Table 3 Local species turnover between 1952 and 2013.** Values ± SE. Estimates represent the probability that a randomly selected species was "new" to the species pool during the modern survey period. Standard errors were derived from nonparametric bootstrapping methods.

| Site | All detections (%) | Visual only (%) |
|---|---|---|
| Overall | 39.3 ± 6.2 | 48.3 ± 6.7 |
| Coniferous | 55.7 ± 10.7 | 68.3 ± 3.5 |
| Marsh | 63.6 ± 12.3 | 64.0 ± 7.3 |
| Mixed deciduous | 58.1 ± 5.0 | 59.1 ± 4.5 |
| Oak woodland | 70.2 ± 6.5 | 75.2 ± 3.0 |
| Willamette | 58.9 ± 8.1 | 54.1 ± 6.9 |

diversity to near or below historic values for all sites. This is notable, considering richness was still higher in the modern era after removing non-visual detections.

Beta diversity, a measure of community dissimilarity among sites, should remain constant with stability and turnover, and decrease with diversity decay. Accounting for differences in species richness, mean beta diversity for the historic period was 0.61, while mean beta diversity for the modern period was 0.18 (0.19 visual only). While modern sites were more similar in community composition than their historic counterparts, beta diversity did not significantly differ between years (all detection types: $t_3 = 2.2$, $p = 0.12$; visual detections only: $t_3 = 1.9$, $p = 0.16$). Both survey periods showed less among-site similarity than expected by chance.

We evaluated patterns of community turnover by quantifying the probability of species replacement between survey periods. Species turnover between 1952 and 2013 was high (Table 3). After removing rarities, only 48 species out of 102 species (39 out of 91 species, visual detections only) were present for both survey periods (Tables S1 and S2). Turnover at individual sites was higher than turnover across the study area. The oak woodland site had the highest species turnover. Removing non-visual detections generally increased turnover values, with the exception of the Willamette River site.

We compared local community results to regional community change using individual-based rarefied BBS data for 10 selected routes with similar habitat. After accounting for

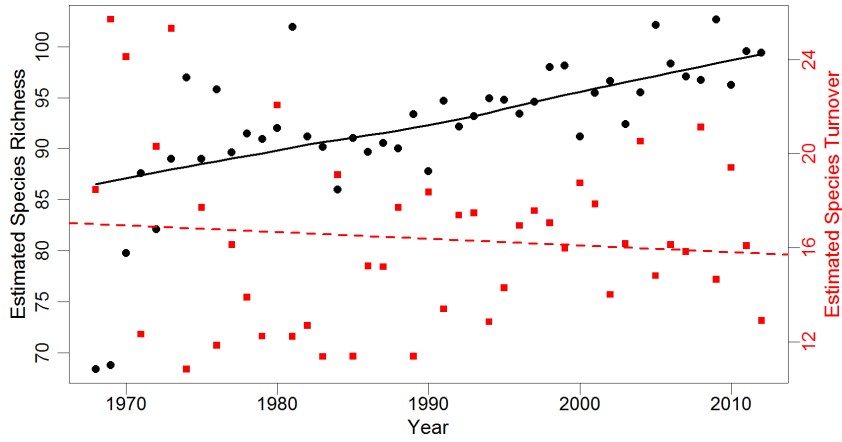

**Figure 1 Estimates of regional species richness and turnover between 1968 and 2012.** Mean values of richness (number of species) and turnover (probability of a species being replaced) for the regional avian community from 1,000 iterations of individual-based rarefaction ($n = 825$) for each year. Species richness estimates are represented by black dots, while turnover estimates are represented by red squares. Black line is a smoothed loess fitted to richness data to represent trend. Red dashed line is a simple linear regression of species turnover ($R^2 = 0.01$). Regional data obtained from 10 Willamette and Umpqua Valley Breeding Bird Survey routes geographically similar to the local study area.

differences in annual survey effort, trends in regional species richness agreed with local results (Fig. 1). Species richness across rarefied samples significantly increased from an estimated 69 species in 1968 (SE = 10.1) to 97 species in 2012 (SE = 10.6). The highest annual estimated species richness was 103 species (SE = 10.8) in 2005. The estimate of overall regional turnover was similar to local turnover, with a 41% probability of species replacement over three decades (SE = 7.7). Mean annual turnover over the 34-year period was 16%. Estimated annual turnover ranged from 11% (multiple years) to 26% (between 1968 and 1969). There was a significant increasing trend in species richness over time after accounting for serial correlation. Regional species richness increased approximately 0.29 species per year between 1968 and 2012 ($p < 0.001$, 95% CI [0.13–0.45]). Regional turnover did not exhibit serial correlation, and had a slight but non-significant decrease of 0.02% over time ($p = 0.51$, 95% CI [−0.11–0.06]; Fig. 1).

## Abundance comparisons

Some patterns of community change predict avian abundances to decrease from urbanization or habitat loss. Considering all detection types, the majority of species in this study were increasing or strongly increasing (40 and 17 species, respectively; Table S3). Another 17 species did not change in categorical abundance over time. 23 species were decreasing in abundance. However, after removing non-visual detections, more birds were decreasing or strongly decreasing in abundance (34 and 8 species, respectively). Only 38 visually detected species increased in abundance to any degree. 11 species did not change in categorical abundance based on visual detections.

The distribution of species among abundance categories shifted significantly over time (all detections: $\chi^2_4 = 10.35, p = 0.04$; visual detections only: $\chi^2_4 = 9.54, p = 0.05$). Species

were more evenly distributed among abundance categories historically. Modern abundances were less evenly distributed and had more species in the "rare" and "abundant" categories. There was variation in the degree to which abundance distributions changed for each individual site. The coniferous and oak woodland site both showed significant shifts in categorical avian abundances regardless of detection type (coniferous site: all detections $p = 0.03$, visual detections $p = 0.002$; oak woodland site: all detections $p = 0.03$, visual detections $p < 0.001$). The coniferous site increased in abundant species, while the oak woodland site increased in rare species. The Willamette River site increased in both abundant and rare species, with a loss in common species; this shift was significant when all detection types were considered ($p = 0.007$).

## DISCUSSION

In 1952, Richard Eddy conducted detailed avian surveys at 6 sites in Benton County, Oregon. We resurveyed 5 of those sites six decades later to evaluate how and to what extent avian community composition changed. Species turnover was high, and there was strong evidence nearly half of the species detected during modern surveys were not present historically. On a local scale, richness increased. Similar levels of turnover and increasing species richness were evident on a regional extent. Our results paint a picture of dynamic species richness, abundances, and overall community change.

Many challenges accompany the use of historic datasets (*Igl & Johnson, 2005*; *Tingley & Beissinger, 2009*). We attempted to account for as many of these as possible through survey methods and analyses. Our greatest challenge was lack of clarity regarding Eddy's detection method. We suspect his dataset contains only visual detections. Most modern surveys emphasize auditory detections, particularly in wooded habitats. Removing non-visually detected species from our surveys provided a conservative estimate of modern avian diversity but improved our ability to compare with Eddy's results. We found historic and modern survey efforts encountered species at similar rates and with similar thoroughness (*Curtis, 2015*). There was no evidence omitting post-breeding season (late July and August) surveys from the historic dataset significantly influenced our conclusions (*Curtis, 2015*). We addressed the issue of uncertain site placement by comparing diversity within sites to adjacent areas. There was no evidence that changing site placement influenced results of modern surveys (*Curtis, 2015*).

Although we addressed potential issues confounding our comparisons with Eddy's surveys, the possibility remains that some changes were the result of methodological differences or changes in detectability, rather than actual ecological changes. While insufficient historic vegetation data and limitations associated with the interpretation of Eddy's data limit the precision of conclusions, we found strong indications of community change during the last six decades.

### Changes in richness and diversity

Species richness increased over time at 4 of the 5 study sites. Local and regional estimates of "true" species richness paralleled these observations. Though richness increased, evenness and diversity both decreased, in some cases to a significant degree. Historic communities

had more species of intermediate abundance (common, uncommon), while modern species tended to be either rare or abundant. This suggests higher species richness may be associated with a decrease in community diversity when abundances vary over time (*La Sorte & Boecklen, 2005*). Simpson's index of diversity is sensitive to changes in evenness. An uneven distribution of individual species abundances should result in lower calculated species diversity, even if species richness increased over the same period. Additionally, variations in evenness may have contributed to overall community instability over the past six decades (*Mikkelson et al., 2011*).

In this study, beta diversity did not significantly change spatially. However, temporal trends in beta diversity were mixed. Other long-term comparisons of avian communities observed temporal declines in beta diversity (*Catterall et al., 2010*; *Shultz, Tingley & Bowie, 2012*; *Davey et al., 2013*). Increases in species richness are frequently associated with decreases in beta diversity because larger species pools share more species between sites (*Davey et al., 2013*). After removing non-visual detections, beta diversity did slightly decrease between survey periods. We detected a few species that colonized all sites since 1952, which contributed to this decline. However, the slight decline in beta diversity was also attributed to encountering the same non-visually detected species across sites during the modern surveys. If the same species (e.g., species whose behavior or physical characteristics made them difficult to see) consistently failed to be visually observed by us at every site, then their removal should increase the heterogeneity of the remaining visually detected species across sites.

## Structural and climatic influences

We identified some degree of vegetation change during this study not characterized by changes in percent cover of habitat types. Variation in vegetation structure and volume may explain, to some extent, the observed changes in species assemblages and abundances (*Vale, Parker & Parker, 1982*; *Holmes & Sherry, 1988*; *Seavy & Alexander, 2011*). The oak woodland and Willamette River sites experienced some of the most pronounced changes in vegetation and land use cover. The Willamette River site transitioned from primarily urban-adjacent open grass and wetlands (*Eddy, 1953*), to a dense, closed-canopy floodplain forest. This site also had one of the largest increases in species richness as well as the highest turnover rate. Nearly a third of the oak woodland site is now coniferous forest. An increase in coniferous-associated species at this site, including Pacific-slope Flycatcher (*Empidonax difficilis*) and Pacific Wren (*Troglodytes pacificus*), may be attributed to changes in forest cover type (*Hagar, McComb & Emmingham, 1996*).

At the marsh site, the elimination of grazing and changes in water management provided more standing water for birds including Pied-billed Grebe (*Podilymbus podiceps*), Wood Duck (*Aix sponsa*), and Tree Swallow (*Tachycineta bicolor*). Species historically detected at the coniferous site—including Western Bluebird (*Sialia mexicana*), MacGillivray's Warbler (*Geothlypis tolmiei*), and Ruffed Grouse (*Bonasa umbellus*)—exhibited decreasing abundances following vegetation growth and canopy closure in the area of the site that was formerly burned (*Eddy, 1953*). Though the Western Bluebird and Ruffed Grouse are

increasing regionally (*Sauer et al., 2014*), local declines in these species (Table S1), as well as MacGillivray's Warbler, may be associated with a loss of non-coniferous habitat (*Hagar, 2007*). It may be that some species presences or absences were associated with changes in the surrounding vegetation community, as suggested by previous studies (*Vale, Parker & Parker, 1982*; *Holmes & Sherry, 2001*; *Seavy & Alexander, 2011*), even if the fine details of such change are difficult to see in this study given limited historic vegetation data.

It is estimated temperatures in the Willamette Valley region increased by approximately 1.5 °C over the past century (*Environmental Protection Agency, 2015*). Data from a single Corvallis climate station suggest local temperature trends resembled regional trends (*NOAA National Climatic Data Center, 2015*). However, it was not possible to quantify small-scale climate change for our study sites. Increasing vegetation structure, density, or shade could influence local temperatures independently of regional trends. Rising temperatures or precipitation changes on a regional level may drive species range shifts and alter regional community composition (*Thomas & Lennon, 1999*; *La Sorte & Thompson III, 2007*; *Illán et al., 2014*). Nevertheless, we cannot evaluate rigorously the degree to which climate may have influenced local species assemblages based on comparisons with these opportunistic historic survey data. It is difficult to parse the effects of climate change on the observed bird communities from local, site-specific factors.

## Local and regional population trends

Local community composition is frequently attributed to site-specific characteristics rather than large-scale influences (*Knick & Rotenberry, 2000*; *Rotenberry & Wiens, 2009*). While we detected some vegetation structure change, largely associated with succession after removal of disturbances, overall land use and vegetation cover remained relatively stable across survey sites. This supports the idea that regional community composition is important in structuring local diversity (*Ricklefs, 1987*). A large regional species pool provides a greater assortment of individuals capable of being recruited into local communities (*Brown et al., 2001*), and may explain why regional gamma diversity complemented local changes in richness and species composition.

Overall, local abundance trends reflected regional population trends. Species with strongly decreasing local populations included Chipping sparrow (*Spizella passerina*), House Sparrow (*Passer domesticus*), Nashville Warbler (*Oreothlypis ruficapilla*), and Northern Rough-winged Swallow (*Stelgidopteryx serripennis*). All of these species were common or abundant in 1952 but not detected during modern surveys. Other birds originally detected but not observed during resurveys include Ring-necked Pheasant (*Phasianus colchicus*), Northern Bobwhite (*Colinus virginianus*), and Common Nighthawk (*Chordeiles minor*). Statewide BBS trends for these species are all negative and, in the case of Chipping Sparrow, Northern Bobwhite, and Northern Rough-winged Swallow, population declines are quite strong (*Sauer et al., 2014*).

Many species with strongly increasing populations on a local scale are also increasing regionally (*Sauer et al., 2014*). It has been suggested that as species richness increases, dominant species, or species with proportionally large numbers of individuals, also

increase (*La Sorte & Boecklen, 2005*). In this study, several previously unobserved species now dominate the community with high categorical abundance. New species, including European Starling (*Sturnus vulgaris*), House Finch (*Haemorhous mexicanus*), and Brown-headed Cowbird (*Molothrus ater*), are some of the most common species in the Willamette Valley (*Hennings & Edge, 2003*). Species not detected historically but frequently observed during modern surveys included House Finch, Anna's Hummingbird (*Calypte anna*), and Acorn Woodpecker (*Melanerpes formicivorus*). The remaining new species, such as Black Phoebe (*Sayornis nigricans*), Eurasian Collared-Dove (*Streptopelia decaocto*), and Sharp-shinned Hawk (*Accipiter striatus*), tended to be categorically rare, because either they were not established in the area (phoebe and collared-dove), or they may now be easier to detect than historically (hawk).

## Patterns of community change

Avian community diversity at 5 sites in the Willamette Valley, Oregon changed over the course of 60 years, but the nature of this change was complex and not easily characterized. Our results are most consistent with community turnover. Increases in species richness contrasted decreases in species evenness and diversity, but the largest change was in community composition. We observed less than half of historically occurring species. Turnover estimated over all sites was a conservative measure of assemblage change because species extirpated from one area may still have been detected at another site.

Estimates of turnover at individual sites were even higher; the odds of species persisting within the local assemblage at any given site were between 25 and 40%. These values agree with those from other research. After 50 years of study, *Parody, Cuthbert & Decker (2001)* reported only 30% of species were consistently present over time within communities. *Diamond (1969)* found species turnover rates of 50–60% after nearly half a century of community change. Likewise both *Catterall et al. (2010)* and *Shultz, Tingley & Bowie (2012)* determined turnover was the driving force of community change over time, as neither species richness nor diversity significantly differed between years. Across the globe, communities may undergo significant assemblage changes without systematic loss of diversity (*Dornelas et al., 2014*).

There was some evidence for a pattern of invasion and homogenization from our results. Most species observed as increasing or strongly increasing in abundance did so across all sites. Newly colonizing species may have contributed to skewed abundances by increasing numbers in both the rare and abundant categories. Immigrant species such as European Starling and Brown-headed Cowbird were observed at nearly every site. The result of this widespread species influx may be the observed increases in richness and slight decrease in beta diversity over time. Further monitoring is needed to evaluate whether new species invasions will ultimately reduce functional and beta diversity as some models predict (*McKinney & Lockwood, 1999*; *Devictor et al., 2007*; *Davey et al., 2013*)

## CONCLUSION

Novel communities are expected as species distributions adjust to changing environmental conditions (*Thomas & Lennon, 1999*; *Williams & Jackson, 2007*). There is concern the

resulting communities will possess altered ecosystem functionality and challenge species' abilities to adapt (*Stralberg et al., 2009*). An underlying implication is that communities would otherwise remain static, or that modern species assemblages are comparatively more natural than those resulting from climate change and anthropogenic disturbance. However, it is difficult to say how much species assemblage variation could be expected even under "normal" conditions (*Magurran et al., 2010*). Our study suggests communities are in a state of flux and "re-shuffle" over decadal periods even with little macro-scale habitat change.

The community turnover observed in this study demonstrates long-term variability of species composition. Drivers of community change may not always relate to climate, vegetation, or human disturbance exclusively. Much has been said about the influence of site-specific conditions on observed community composition (*Knick & Rotenberry, 2000*; *Rotenberry & Wiens, 2009*). However, our results demonstrate the association between regional communities and community change at smaller scales. Community turnover is an ecologically important pattern useful for understanding the underlying mechanisms structuring communities (*Chase et al., 2011*). Assemblage flux may contribute to community instability, even when richness remains constant or increases, in cases where the addition of new species reduces evenness. Population instability resulting from reduced evenness (*Mikkelson et al., 2011*) may drive other local species to disappear, producing the observed community turnover.

Despite the challenges our comparison faced, we identified significant species assemblage changes. Richard Eddy's (*1953*) dataset is the only one of such detail currently known for the Pacific Northwest from that era, and provides us with a unique look at historic avian communities. Some may consider the analytical challenges presented by historic datasets to be insurmountable. However, to discard past data because it no longer meets modern requirements is to ignore a valuable perspective on previous conditions. Historic data remind us that biological communities are in flux and may not be easily characterized by a few seasons of data collection. As more researchers seek ways to preserve biodiversity in the face of global climate change, historic datasets present an essential perspective on how community diversity varies over time.

## ACKNOWLEDGEMENTS

We gratefully acknowledge Stuart Pimm, Gregory Mikkelson, Terry Root, and one anonymous reviewer who provided valuable comments and suggestions for earlier drafts of this manuscript. We thank John Alexander, Pat Kennedy and Jim Peterson for their advice during the design, analysis, and writing process. John Van Sickle and Bruce McCune provided statistical advice. Shelley Hansen allowed access to the mixed deciduous site through the OSU poultry facilities. Sharon Smythe, Tyler Hallman and Chad Marks-Fife accompanied and documented field surveys.

### Funding

This research was funded by Oregon State University, with support from the Bob and Phyllis Mace Watchable Wildlife Professorship and a scholarship from the Santiam Fish and Game Association. The funders had no role in study design, data collection and analysis, decision to publish, or preparation of the manuscript.

### Grant Disclosures

The following grant information was disclosed by the authors:
Oregon State University.
Bob and Phyllis Mace Watchable Wildlife Professorship.
Santiam Fish and Game Association.

### Competing Interests

The authors declare there are no competing interests.

### Author Contributions

- Jenna R. Curtis conceived and designed the experiments, performed the experiments, analyzed the data, contributed reagents/materials/analysis tools, wrote the paper, prepared figures and/or tables, reviewed drafts of the paper.
- W. Douglas Robinson conceived and designed the experiments, contributed reagents/materials/analysis tools, wrote the paper, reviewed drafts of the paper.

### Data Availability

The following information was supplied regarding the deposition of related data:
Breeding Bird Survey Data (for regional dataset): Pardiek et al., 2014. (https://www.pwrc.usgs.gov/bbs/RawData/)
GAP Analysis Data (for land use and vegetation cover): *US Geological Survey, 2012*. (http://gapanalysis.usgs.gov/gaplandcover/data/download/)
Historic Dataset: Eddy R. 1953. "Summer bird habitats in the Corvallis area, Willamette Valley, Oregon". Master's Thesis: Corvallis, Oregon. (OCLC number 18957033).

### Supplemental Information

Supplemental information for this article can be found online at http://dx.doi.org/10.7717/peerj.1152#supplemental-information.

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
