# Peer review of "Sixty years of change in avian communities of the Pacific Northwest"

_PeerJ, doi:10.7717/peerj.1152_

## Round 0.1 · original submission · Major Revisions

You are fortunate to have three distinguished reviewers and to have had them provide such explicit recommendations. I urge you to make your changes promptly, as that enables both me and the reviewers to have these reviews fresh in our minds when we see your revisions and the responses to comments.

·

Basic reporting

I think the paper falls short with regard to "sufficient introduction and background to demonstrate how the work fits into the broader field of knowledge".

One part of this problem is that the paper provides a weak theoretical basis and rationale for the study. Where do the three hypotheses of "stability, diversity decay, and turnover" come from? Furthermore, "stability" as interpreted in the paper is a straw man. Who, or what theory, ever predicted zero change in community composition? Perhaps an article cited in another paper I'm currently reviewing could help the authors think about how to sharpen up this aspect of their manuscript: Scheiner, S. M. 2013. The ecological literature, an idea-free distribution. Ecology Letters 16:1421-1423.

Another part of the same problem – unfortunately also quite common in ecology – is at least one instance of spurious citation. The authors cite Mikkelson et al. (2011) in support of the assertion that "Community turnover is an ecologically important pattern useful for understanding the underlying mechanisms structuring communities". But that paper does not even use the word "turnover", and instead examines relations between species diversity and the stability of abundance (not community composition).

Experimental design

The methods employed for estimating diversity seem up-to-date and rigorous, though I am not familiar enough with some of them to assess their validity or appropriateness. I recommend additional commentary - and a closer look at whether some of them are necessary - to save other readers from encountering the same problem.

Validity of the findings

No comments

Reviewer 2 ·

Basic reporting

Basic reporting and format is ok. Suggestions for improvement:

1) Tables 2 and 3 have the sites and other information (years, etc.) as columns, whereas Table 1 has the sites as columns. For clarity, I suggest making Table 1 match the others. Also in Table 1, the Chao 95% CIs are in separate rows. In the other tables, CI values are presented with the main value like this XX (YY - ZZ). I would do the same, again for consistency.

2) Figure 1 should have larger text for ticks and and axis labels, and thicker lines for the fits (but see comment on fits below).

Line 111: "Without NON-visual", surely?

Experimental design

Generally speaking the design is ok. One technical mistake — fitting a simple regression to time-series data, as per Figure 1, is definitely not ok, as non-independence of points means significance will be overestimated. Assuming equally spaced points and first-order dependence (no long-term cycling), a simple approach is to difference the time series data, and then perform a t-test on the set of differences to see if the mean is significantly different from zero. (The mean of the differences will be the same as the fitted slope from the earlier method.) Even slightly better would be to log-transform the richness values before differencing, as this corresponds to an exponential trend rather than a linear one, better for richness data that must be bounded at zero. (In practice there will be little difference in these data, where all the richnesses are well away from zero.) See any number of papers on time series analysis for details.

Validity of the findings

This paper is a valiant effort to make the most out of an old dataset that was collected using only partly-understood and clearly less-than-ideal methods. I do believe the effort is worthwhile, as historical datasets are relatively rare and provide important insight into long-term trends. The authors have done right thing with respect to the biggest unknown, which is how Eddy actually performed his survey, by doing their modern version two ways that almost certainly 'bookend' his. And the analyses themselves (with the exception of the time-series regression mentioned above) seem well done.

I'm less enthused about the 'framing device' of the "three patterns of community change," presented as alternative hypotheses. I've worked in this field, and they are not something I immediately recognize, and I don't know that anyone would really think of them as alternatives. Most classic papers acknowledge the simultaneous influence of diversity/richness change and turnover, and the 'stability' hypothesis almost qualifies as a straw man at this point. In fact the patterns discovered in this paper fit very nicely a fourth, well-studied model: 'invasion/homogenization'. In that model, local richness increases because of invaders, abundance distributions tend to become more skewed, as a result, diversity measures that include evenness can go either way, and beta diversity decreases because the same common invaders (starlings, cowbirds, etc.) invade a variety of sites. It's not a perfect fit, but it's close, and having this upfront, rather than the odd alternatives presented, would make for a more compelling structure to the paper. For example, the 'lower beta diversity' trend was found in the paper (albeit not significantly), but attributed in the Discussion solely, and rather obscurely, to the methodological problem of visual vs. auditory identifications, whereas it might have been placed in the invasion/homogenization context.

So overall, I recommend a rethinking of the Introduction along these lines, and I think this may well make the Results-Discussion text flow a little more easily and present a more compelling paper overall.

·

Basic reporting

In my opinion, not enough information has been given to the reader. I would have like to know which species were found in 2013 and not in 1952, and the other way around. This would need to be done by habitat type. Also the choice of sites were not completely clear.

Experimental design

The type of censusing technique varied between the 2 time periods. This really was not addressed. I know that doing transects usually records more species and individuals than point counts do, but this was not addressed.

Validity of the findings

The possible changes due to climate disruption are completely ignored.

Additional comments

60 years of change in avian communities of the Pacific Northwest. By J. R. Curtis and W. D. Robinson
Submitted to PeerJ

This article, which examines the changes in birds over 60 years in the Willamette Valley, OR, is well written. As expected a fairly large number of changes in the birds have occurred over the 60 years. The authors looked at 5 of the 6 sites censuses 60 years prior. They found that at 4 of the 5 sites species richness increased, while the evenness at the sites decreased with time. Beta diversity did not change over time, but community diversity did change, yet the pattern is not clear over all sites.

I applaud the authors in using historic data to help us understand what has changed over a long temporal scale. Making comparisons over such a long time scale are difficult and can be problematic. As the authors have done, however, many of these problems can be addressed and many of those that cannot, they explain them and why they cannot be addressed.

The statistical analyses for this paper are extensive and to my understanding appropriate.

I do have concerns about the paper:
Only 5 of the 6 original sites were censused, however, 2 of these 5 may not have been where the original sites were. They were “…’likely areas” adjacent to the selected survey areas.” I am not clear what the part of the sentence means that I have indicated in bold (line 63). I also do not understand what is meant by (lines 64 & 65) “We compared bird abundances in the 2 sites to adjacent “likely areas” to confirm site placement did not affect survey results.” What kind of comparison was made, to what (2 sites to adjacent ‘likely areas’?), and how was it confirmed?

The type of censusing technique varied between the 2 time periods. The differences caused by this difference was not addressed. I know that doing transects usually records more species and individuals than point counts do, but I am not sure how censusing while walking an area for 2 hours compares to point counts. Does anyone? I assume someone does.

The original censuses were done June through August. Because the authors were interested in looking at changes in “…historic breeding season bird communities…” (line 73) they censused (line 69) from mid-May through 1st week of July, and only looked at the historical data recorded during the (line 74) “…first 5 visits to each site.” There is a 2 week shift from beginning of June to mid-May? Why 2 weeks? Many breeding birds are arriving earlier in spring than they did in the 1960s. There are many articles on changing phenology primarily due to climate change. There is no indication that these studies were examined and used to justify the 2-week earlier starting period. I am not sure that 2 weeks is the correct timing to use when making such a change in the timing of censusing. If there has been roughly a 5 days/decade earlier shift from mid-1960 to mid-1990—as can be found in the climate change literature—then a 2-week shift is not appropriate, but this is never discussed. (In fact, climate change is not discussed until line 409.) Also, we are not told how the historic data were collected (and need to be). I assume that Eddy rotated his censusing locations so that the starting times for all 5 sites used were in the first week of June, but this needs to be made explicit.

After the recent censuses were completed, the authors noticed that birds only heard were probably not included in the historic census They were included in the recent census and no effort was made to see these heard-only birds. The authors tried to account for this by dropping out the heard-only species, but as the authors indicated, dropping these species could easily have biased the data. This is true and actually pretty easy to quantify by going out and doing censuses (does not need to be in the exact locations, just in similar habitat) to calibrate the amount of bias dropping heard-only species that probably could have been seen if the effort was made. Indeed, the removal of heard-only birds greatly changed the number of decreasing and increasing species (line 261).

The authors list the species in Table S1 that did not change in abundance, increased, decreased, increased strongly or decreased strongly. However, I could not find the names of the birds that were found in one time period and not in the other time period. The difference shown in Table 1 are pretty dramatic. I am surprised that these species were not discussed. Climate change is known to have caused species, including birds, to move north in the Northern Hemisphere, following the warming temperatures. Undoubtedly this is occurring among the species examined. Possible examples of birds expanding with warming are: Black Phoebe, and Anna’s Hummingbird. I don’t know if climate change has caused these species apparently to move into the valley, but looking at the range maps in a field guide published in 1960 and one published in 2000, the possibility exists, yet no mention was made of this. Indeed, it seems as though the authors were completely avoiding even the mention of climate change. This phrase only occurs once, as does the word climate, and that is in the Conclusion, where (lines 402-404) they say, “An underlying implication is that communities would otherwise remain static, or that modern species assemblages are comparatively more natural than those resulting from climate change and anthropogenic disturbance.” They go on to say, “However, it is difficult to say how much species assemblage variation could be expected even under “normal” conditions (Magurran et al., 2010). Our study suggests communities are in a state of flux and “re-shuffle” over decadal periods even with little macro-scale habitat change.” Besides implying (actually more than implying) that climate change is not an anthropogenic disturbance, they are saying that the communities of birds examined are changing even though there was little change in habitat. I hazard a guess that the temperatures at these location have changed significantly, which I would not say was “….little macro-scale habitat change.”

So the bottom line is that the authors’ intention of trying to determine and understand change in bird communities over a 60-year period were great, but they seem to miss the mark because of the several reasons indicated above. None of the issues is fatal, but in my opinion each needs to be addressed and the manuscript reassessed.

---

## Round 0.2 · Minor Revisions

Thanks for resubmitting. Please address the reviewers' comments in a revised version of your manuscript.

·

Basic reporting

Mostly well-reported.

However, framing the questions in terms of highly artificial straw men with no credible theoretical basis (e.g., "stability" i.e. no turnover or diversity change whatsoever, "turnover" but no change at all in diversity) still detracts from that. Why not simply state that you measured changes in species richness, species evenness, and species composition as potentially independently varying quantities? Your results concerning these community characteristics are interesting enough in themselves.
Fixing this would mean, e.g., simply dropping all of Paragraph 2, and several other unneeded sentences throughout the text.

Also, the explanation of Simpson's diversity is badly muddled. You should simply say that Simpson's index of diversity is the reciprocal of the chance that two randomly-sampled individuals are of the same species.

Finally, after reading this second submission I do not understand the figure.

Experimental design

Lots of good rigorous work

Validity of the findings

To improve robustness, you should check whether Simpson's evenness (Simpson's diversity divided by species richness) agrees with your measure of evenness based on abundance categories. See Mikkelson et al. (2011) for comments and a reference justifying the use of Simpson's evenness, as opposed to some other popular metrics.

Reviewer 2 ·

Basic reporting

Issues fixed.

Experimental design

Glad to see that temporal trends are being calculated more appropriately.

Validity of the findings

The manuscript is certainly better. I still have some issues with the use of ‘stability’ even as a null model: it’s almost like a modern paper on geology feeling the need to mention uniformitarianism as a null model! Ok, not quite, but essentially *all* the evidence is that at small scales there is significant turnover. But if they want to use stability as a point of departure, and are clear that it’s not anything that would be expected (esp. with birds), I’m ok with it.

Additional comments

A different reviewer suggested a look at climate change as a possible explanation for changes in community composition, and the authors duly calculated trends in temperature and precipitation from the nearest suitable weather station. Obviously that reviewer should weigh in on this, but I can’t help notice that according the new manuscript, there was ‘little to no evidence that local climate changed over time’ (line 417). The strongest, and only significant, trend was a 1.8 degree C increase in July temperature over 61 years. However, first note that the EPA-calculated increase for Western Oregon generally is 1.5-2.0 degrees F per century!

http://www.epa.gov/climatechange/science/indicators/weather-climate/temperature.html, figure 3.

This is 0.015-0.02 degrees F per year, and the authors find 0.01-0.03 degrees C per year in their data, which various from being in line with, or in the case of July, up to 3-4 times higher than, the EPA regional trends. Ah, you say, none except July were significant! Doing one’s own analyses on data from a single local station is excessively conservative. A Bayesian would point out politely that all the accumulated evidence for regional climate change provides a very strong ‘prior.’

There is no question that there is climate change happening here as it is almost everywhere, and the local data actually confirm that it is happening at the same or greater rate than in the region more generally. I strongly suspect that the intent of the other reviewer was not to have the authors conduct their own analysis, but rather to address the possible types of impact that climate change may have on these avian communities. The authors say that this is ‘outside the scope of our paper.’ Maybe so — my comment here is simply that to the extent that local temperature data are addressed in the paper, I don’t think the 'little to no evidence’ conclusion is fair. Better to say that local data matched or exceeded regional trends *but* separating climate change from other processes driving community change, like changes in grazing patterns and water management, is very difficult, especially with opportunistic historical data.

OVERALL: I think the MS is improved enough to cross the barrier of being suitable for publication in PeerJ., but the climate change section still needs some minor revision.

---

## Round 0.3 · accepted · Accept

Thank you for taking care of the changes required by the reviewers.

On reading through the manuscript and supplementary materials, I found a few grammatical problems — so check the red and green underlines (in Word) to ensure that what gets posted online will be correct.